



# Comparison of scanning aerosol LIDAR and *in-situ* measurements of aerosol physical properties and boundary layer heights

Hengheng Zhang[1], Christian Rolf[2], Ralf Tillmann[3], Christian Wesolek[3], Frank Gunther Wienhold[4], Thomas Leisner[1], and Harald Saathoff[1]

[1]Institute of Meteorology and Climate Research, Karlsruhe Institute of Technology, Eggenstein-Leopoldshafen, Karlsruhe, Germany
[2]Institute of Energy and Climate Research - Stratosphere (IEK-7), Reseach Center Jülich, Wilhelm-Johnen-Straße, Jülich, Germany
[3]Institute of Energy and Climate Research - Troposphere (IEK-8), Reseach Center Jülich, Wilhelm-Johnen-Straße, Jülich, Germany
[4]Institute for Atmospheric and Climate Science, ETH Zürich, Universitätstrasse 16, Zürich, Switzerland

**Correspondence:** Hengheng Zhang (hengheng.zhang@kit.edu) and Harald Saathoff (Harald.Saathoff@kit.edu)

**Abstract.** The spatial-temporal distribution of aerosol particles in the atmosphere has a great impact on radiative transfer, clouds, and air quality. Modern remote sensing methods as well as airborne *in-situ* measurements by unmanned aerial vehicles (UAV) or balloons are suitable tools to improve our understanding of the role of aerosol particles in the atmosphere. To validate the measurement capabilities of three relatively new measurement systems and to bridge the gaps that are often encountered

between remote sensing and *in-situ* observation as well as to investigate aerosol particles in and above the boundary layer, we conducted two measurement campaigns and collected a comprehensive dataset employing a scanning aerosol LIDAR, a balloon-borne radiosonde with the Compact Optical Backscatter Aerosol Detector (COBALD), an optical particle counter (OPC) on a UAV, as well as a comprehensive set of ground-based instruments. The extinction coefficients calculated from near-ground-level aerosol size distributions measured *in-situ* are well correlated with those retrieved from LIDAR measurements

with a slope of $1.037 \pm 0.015$ and a Pearson correlation coefficient of 0.878, respectively. Vertical profiles measured by an OPC-N3 on a UAV show similar vertical particle distributions and boundary layer heights as LIDAR measurements. However, the sensor, OPC-N3, shows a larger variability in aerosol backscatter coefficient measurements with a Pearson correlation coefficient of only 0.241. In contrast, the COBALD data from a balloon flight are well correlated with LIDAR-derived backscatter data from the near ground level up to the stratosphere with a slope of $1.063 \pm 0.016$ and a Pearson correlation coefficient of

0.925, respectively. This consistency between LIDAR and COBALD data reflects a good data quality of both methods and proves that LIDAR can provide reliable and spatial distributions of aerosol particles with high spatial and temporal resolutions. This study shows that the scanning LIDAR has the capability to retrieve backscatter coefficients near ground level (from 25 m to 50 m above ground level) when it conducts horizontal measurement which isn't possible for vertically pointing LIDAR. These near-ground-level retrievals compare well with ground-level *in-situ* measurements. In addition, *in-situ* measurements on

the balloon and UAV validated scanning LIDAR retrievals within and above the boundary layer. The scanning aerosol LIDAR allows us to measure aerosol particle distributions and profiles from the ground level to the stratosphere with an accuracy equal or better than *in-situ* measurements and with a similar spatial resolution.



## 1 Introduction

The large varieties of aerosol spatial-temporal distributions in the atmosphere cause large uncertainties in radiative forcing
globally (Ramanathan et al., 2001) and these uncertainties have a great impact on climate change (Stocker, 2014). The distributions and evolution of aerosol are related to the emission of aerosols (Grythe et al., 2014; Tegen and Schepanski, 2018; Hamilton et al., 2022) and the their loss pathway (Poreh and Cermak, 1964; Cheng et al., 2011; Xiang et al., 2019; Xue et al., 2022). In addition, another important factor affecting radiative forcing is aerosol optical properties (e.g. single scatter albedo (SSA), LIDAR ratio, scatter and absorption coefficients) (Alam et al., 2011; Romshoo et al., 2021), which also have large
varieties for different types of aerosols (Lesins et al., 2002; Floutsi et al., 2022).

Many methods have been used to measure the spatial-temporal distribution and aerosol optical parameters regionally and globally. One of the most successful instruments for this purpose is the Moderate Resolution Imaging Spectroradiometer (MODIS) on Terra and Aqua satellites (Filonchyk and Hurynovich, 2020; Qin et al., 2021). MODIS can provide column-integrated optical parameters like aerosol optical depth (AOD), Ångström exponent (AE), and single scatter albedo (SSA) to study the optical
properties of mineral dust (Kaufman et al., 2005; Ginoux et al., 2012), urban aerosol (More et al., 2013; Munchak et al., 2013), forest fire smoke (MAE, 2009; Huesca et al., 2009) etc. Another successful satellite mission is the Cloud-Aerosol LIDAR and Infrared Pathfinder Satellite Observations (CALIPSO). CALIPSO combines an active LIDAR instrument with passive infrared and visible imagers to probe the vertical structure and properties of thin clouds and aerosols over the globe (Winker et al., 2009; Wang et al., 2021; Salehi et al., 2021).

In addition to these satellite missions, ground-based remote sensing methods are used to investigate aerosol optical properties (Kotthaus et al., 2023). Over the last decades, many ground-based observation networks were established to investigate aerosol properties regionally and globally. For example, the AERONET (AErosol RObotic NETwork) project is a federation of ground-based remote sensing aerosol networks that provides globally distributed observations of spectral aerosol optical depth (AOD), inversion products, and precipitation water in diverse aerosol regimes (Holben et al., 1998; Prasad and Singh, 2007;
Mielonen et al., 2009). The Micro-Pulse LIDAR Network (MPLNET) is a federated network of Micro-Pulse LIDAR (MPL) systems designed to measure aerosol and cloud vertical structure, and boundary layer heights (Welton et al., 2006; Lolli et al., 2018). The European Aerosol Research LIDAR Network (EARLINET) is a multi-wavelength LIDAR network designed to create a quantitative, comprehensive, and statistically significant database for the horizontal, vertical, and temporal distribution of aerosols on a continental scale (Pappalardo et al., 2014a; Marinou et al., 2017).

*In-situ* measurements can also help us better understand aerosol optical properties. The most common instruments are the nephelometer and aethalometer, which can measure the wavelength-dependent optical parameters like scatter and absorption coefficients of aerosol particles (Anderson et al., 1996; Drinovec et al., 2015). The aerosol optical parameters are determined by particle size distribution, particle shape, and complex refractive index (Bohren and Huffman, 2008; Yao et al., 2022). The size distribution can be measured by different kinds of particle sizers like Scanning Mobility Particle Sizer (SMPS), Optical
Particle Counter (OPC), and Aerodynamic Particle Sizer (APS). The aerosol complex refractive index is related to the aerosol chemical composition which can be measured by aerosol mass spectrometry. For decades, these *in-situ* aerosol characterization



instruments not only provided valuable datasets at ground level (Huang et al., 2019; Jiang et al., 2022) but also were deployed on aircraft, balloons, and unmanned aerial vehicles to get vertical profiles of aerosol parameters (Bahreini et al., 2003; Zhen et al., 2018; Brunamonti et al., 2021).

LIDAR is a powerful tool to measure the spatial distribution and optical parameters of aerosol (Böckmann et al., 2004; Matthais et al., 2004). Although many results have reported aerosol measurements by LIDAR (Matthias and Bösenberg, 2002; Pappalardo et al., 2014b; Hofer et al., 2020; Ceolato and Berg, 2021), there are fewer reports on comparison of *in-situ* measurement with LIDAR measurement to quantify uncertainties of LIDAR retrievals (Düsing et al., 2018; Xiafukaiti et al., 2020; Düsing et al., 2018). In addition, most vertical pointing LIDAR systems have overlap gap between the detector's field of view and

the laser beam from tens to around one thousand meters, which makes it difficult to get valid measurement near the surface (Wandinger and Ansmann, 2002) to compare with ground level *in-situ* measurements. However, scanning LIDAR can conduct horizontal measurements allowing to get vertical profiles of aerosol particles and boundary layer structure near the ground level (Althausen et al., 2000). In addition, scanning aerosol LIDAR can also determine LIDAR ratios to reduce the uncertainties in the LIDAR retrievals (Fernald, 1984; Zhang et al., 2022).

In recent years, vertical profiles of aerosol are also investigated more and more by Unmanned Aerial Vehicles (UAV) and LIDAR. For example, Liu et al. (2020) used the UAVs and LIDAR to study the vertical distribution of $PM_{2.5}$ and interactions with the atmospheric boundary layer during the development of heavy haze pollution. Ferrero et al. (2019) compared the backscatter coefficient retrieved from LIDAR with that calculated from aerosol size distributions measured by OPC on tethered balloons in the Arctic to study the role of aerosol chemistry and dust composition in a closure experiment. Zhang et al. (2021)

compared boundary layer heights retrieved from aerosol LIDAR and tethered balloon measurements in semi-arid regions. Liu et al. (2021) found that wind shear generating turbulence reshaped the vertical profiles of parameters such as potential temperature ($\theta$) and $PM_{2.5}$ in the nocturnal boundary layer, which was the key factor leading to the development of entrainment at nighttime. Reineman et al. (2016) used ship-launched fixed-Wing UAVs to measure the marine atmospheric boundary layer and ocean surface processes. In addition, the vertical profiles of atmospheric parameters related to aerosol process such as tem-

perature (Zarco-Tejada et al., 2012), relative humidity (Spiess et al., 2007), wind (Spiess et al., 2007) and ozone concentration (Guimarães et al., 2019) are also obtained from UAV flights.

However, to our best knowledge, so far no dedicated comparison of scanning LIDAR measurement with *in-situ* observation has been performed over a wide altitude range. Therefore, we compared datasets on aerosol spatial-temporal distributions and evolutions combining remote sensing and *in-situ* measurements. Two field campaigns were conducted employing a scanning

aerosol LIDAR, a radiosonde with a backscatter sensor, an OPC on a UAV, and a comprehensive set of ground-level instruments. The first field campaign was conducted in downtown Stuttgart to compare LIDAR retrievals with ground level *in-situ* measurements. The second field campaign was done at the Jülich research center to compare LIDAR retrievals with OPC measurements on a UAV and a COBALD backscatter sensor on a radiosonde. The aim of this work is to compare the different methods in aerosol measurements, to validate scanning LIDAR retrievals, to discuss the uncertainties of the different methods

and the boundary layer evolutions from LIDAR and UAV retrievals.





## 2   Methods

Two field campaigns were conducted in downtown Stuttgart and at Jülich research center to compare scanning aerosol LIDAR measurements with different *in-situ* measurements. The first field campaign was conducted from February $5^{th}$ to March $5^{th}$, 2018 in downtown Stuttgart (9.2024° E 48.7986° N, 247 m above sea level) employing a mobile container and a scanning aerosol LIDAR on the roof of the container. The ground-level *in-situ* measurements deployed in this mobile container provided aerosol particle size distributions, aerosol chemical composition, and meteorological information (Huang et al., 2019). The second field campaign was conducted from July $5^{th}$ to $12^{th}$, 2018 at Jülich research center (6.4131° E, 50.9084° N, 110 m above sea level) employing a scanning aerosol LIDAR, a COBALD sensor hosted by a Vaisala RS41-SGP radiosonde, and an OPC on UAV. The scanning LIDAR called KASCAL used in these two field campaigns was developed by Raymetrics (LR111-ESS-D200, Raymetrics Inc.). A UAV (eBee, senseFly) carrying one OPC (OPC-N3, Alphasense Inc.), weather sensors and Global Positioning System (GPS) sensors provided altitude-dependent particle size distribution and also meteorological information above the Jülich research center. In addition, atmospheric parameters like pressure, temperature, relative humidity and wind information from the ground to 30 km above Jülich research center were gathered by a GPS-equipped radiosonde onboard a balloon that carried COBALD to measure altitude-dependent *in-situ* backscatter coefficients at two wavelengths (455 nm & 940 nm) (Brunamonti et al., 2021). The measurements during this work indicated that the basckatter was dominated by smaller particles with low depolarisation ratios so that it seemed justified to use a spherical model to represent these aerosol particles (Khlebtsov et al., 2005; Moroz, 2009; Wang et al., 2023). Hence, a Mie code (Leinonen, 2016) was used to calculate extinction coefficients and backscatter coefficients from aerosol size distributions for comparison with the LIDAR retrieval.

### 2.1   Scanning aerosol LIDAR

The 3D scanning LIDAR (KASCAL) used in the above two field campaigns has an emission wavelength of 355 nm and is equipped with elastic, depolarization, and vibrational Raman channels, hence allowing to retrieve extinction coefficients, backscatter coefficients, and depolarization ratios. The laser pulse energy and repetition frequency are 32.1 mJ and 20 Hz, respectively. The laser head, 200 mm telescope, and LIDAR signal detection units are mounted on a rotating platform allowing zenith angles from -7° to 90° and azimuth angles from 0° to 360°. This LIDAR works automatically, time-controlled, and continuously via software developed by Raymetrics. Detailed information can be found at https://www.raymetrics.com/product/3d-scanning-LIDAR, last access: 8 March 2021 (Avdikos, 2015; Zhang et al., 2022). During the first field campaign in downtown Stuttgart, the LIDAR conducted zenith scanns with an elevation angle from 90 ° to 5° in steps of 5°. The measurements at 5° were used over a range representative of an altitude of 25-50 m to compare with ground-level *in-situ* measurements (3.7 m above ground level). It is assumed that these values are comparable within the mixing layer. During the second field campaign at Jülich research center, the LIDAR conducted zenith scans during UAV launch and the measurements at all elevation angles were used to get vertical profiles of aerosols from ground level up to the free troposphere to compare with an OPC measurement on the UAV. In addition, the LIDAR also conducted vertical pointing measurements in the night of July $12^{th}$, 2018 at Jülich research center to compare the vertical profiles of backscatter coefficients from LIDAR retrievals and COBALD measurement



on board of a radiosonde.

For the data analysis and calibration of the LIDAR system, we followed the quality standards of the European Aerosol Research LIDAR Network (EARLINET) (Freudenthaler, 2016). For data analysis of zenith scans, we determine the vertical backscatter coefficient profiles using the Klett-Fernald method (Fernald, 1984; Klett, 1985). And these vertical profiles of aerosol backscatter coefficients was used as the reference values for other observation angles. In addition, the measured temperatures and pressures from UAVs and balloons were used to calculate the molecular backscatter coefficients which can be used in

LIDAR retrievals.

The atmospheric boundary layer height determined from LIDAR by using the Haar wavelet transform (HWT) method is defined as

$$z_{HWT} = \max[w_f(a,b)] = \max \frac{1}{a} \int\limits_{z_{min}}^{z^{max}} X(z) H(\frac{z-b}{a}) \, dz \tag{1}$$

In which $w_f$ is the covariance transform value, $X(z)$ is the range corrected LIDAR signal defined as $X(z) = P(z) * z^2$, and

$H(\frac{z-b}{a})$ is the Harr wavelet function as defined as followed:

$$H(\frac{z-b}{a}) = \begin{cases} 1 & b - \frac{a}{2} \leq z \leq b \\ -1 & b \leq z \leq b + \frac{a}{2} \\ 0 & \text{elsewhere} \end{cases} \tag{2}$$

The dilation a was tested and set to be 75 m for this work. $Z_{min}$ and $z_{max}$ are the lower and upper heights for the LIDAR signal profile, respectively. In addition, the boundary layer height was also retrieved from vertical profile of potential temperature by using gradient method. (Seidel et al., 2010; Li et al., 2021).

## 2.2 Ground-level *in-situ* measurements in downtown Stuttgart

The ground-level *in-situ* instruments were deployed in a mobile container that was deployed in a parks downtown Stuttgart. Ambient temperature, relative humidity, wind direction, wind speed, global radiation, pressure, and precipitation data were measured by a meteorological sensor (WS700, Lufft GmbH). Trace gases ($O_3$, $CO_2$, $NO_2$, $SO_2$) were measured with the gas monitors (Environment S.A). Particle number concentrations were recorded with two CPCs (CPC 3774, 3022, TSI Inc,).

Particle size distributions were measured with SMPS (DMA: TSI 3080, TSI Inc; CPC: CPC3022, TSI Inc), and OPC (Fidas200, Palas, Inc). The OPC (Fidas200, Palas, Inc.) continuously measured particles in the size range of 0.18 - 18 $\mu$m. The OPC used Lorenz-Mie theory to determine the particle number size distribution and this size distribution can be used to calculate extinction coefficients via a Mie code (Leinonen, 2016). In this experiment, Fidas200 was operated with a flow rate of 5 L/min and with a time resolution of 1 s.

Figure 1 shows the workflow in deriving the aerosol extinction coefficients from Mie calculations based on *in-situ* aerosol characterisation instruments. The aerosol sizer (e.g. OPC) can provide dry aerosol particle size distribution, which can be converted to the ambient aerosol size distributions by using hygroscopic growth factors ($\kappa$) calculated from aerosol chemical




composition using the ISORROPIA II thermodynamic equilibrium model (Fountoukis and Nenes, 2007). The aerosol chemical composition was measured by HR-TOF-AMS (Aerodyne Inc.). Most aerosol particles are constrained and well-mixed within
the boundary layer and the aerosol complex refractive index remains almost constant (Raut and Chazette, 2008). Although, the sun photometer is integrating over the whole vertical column, the relatively high aerosol concentrations in the boundary layer dominate (Li et al., 2017). Therefore, it seem justified to use the aerosol complex refractive index derived from a nearby sun photometer (CE-318). Hence, we used the aerosol complex refractive index derived from a nearby sun photometer (CE-318). With ambient aerosol size distribution and complex refractive index, optical parameters (e.g. extinction coefficients) were calculated to compare with LIDAR retrievals.

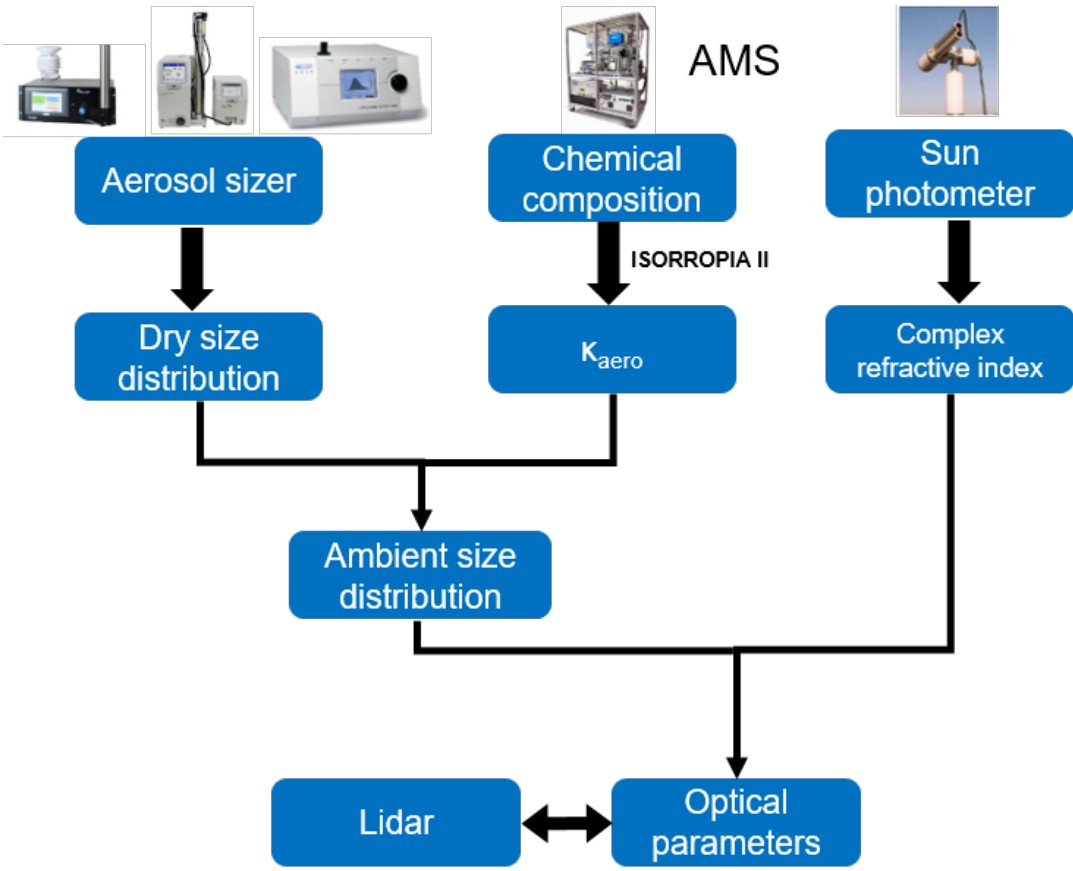

**Figure 1.** Process flow in deriving aerosol extinction coefficients from Mie calculation and parameters used in Mie calculations. $\kappa_{aero}$ is the composition dependent hygroscopicity growth factor.


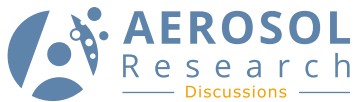

## 2.3 UAV and balloon-borne measurements at Research Center Jülich

Data of an OPC (OPC-N3, Alphasense, Inc) on a UAV and a COBALD backscatter sensor (Institute for Atmospheric and Climate Science, ETH Zurich) on a balloon were collected at Jülich research center in July 2018. The UAV used in this field campaign is a fixed-wing drone (eBee, senseFly) which is operated by the Institute of Energy and Climate Research - Tropo-
sphere (IEK-8). Its payload is 320 g at a total weight of 750 g with the highest observation altitude of approximately 1200 m above ground level. The ascent and descent velocity of this UAV was around 3.2 m/s. The measurement sensors were mounted inside the UAV. The size distributions were measured in real-time with a time resolution of 1.6 s by OPC-N3. Additionally, atmospheric parameters such as air temperature, air pressure, relative humidity, wind speed, and wind direction were measured with a temporal resolution of 1 s. The UAV was launched 5 times during the morning from 7:00 to 10:00 on July $9^{th}$ to measure
the boundary layer dynamics in the early morning and was launched 7 times from 03:50 to 16:30 on July $12^{th}$ to measure the boundary layer transition from nocturnal boundary layer to the mixing layer. The detailed UAV flights information can be found in Table 1.

**Table 1.** Time, altitude, and duration of UAV flights for the experiments on July $9^{th}$ and July $12^{th}$, 2018.

| Flight number | Date | Minimum altitude (m a.s.l.) | Maximum altitude (m a.s.l.) | Duration (s) |
|---|---|---|---|---|
| 2018070901 | 2018.07.09 07:39 | 90.9 | 1246.5 | 709.5 |
| 2018070902 | 2018.07.09 07:48 | 92.4 | 1244.8 | 705.1 |
| 2018070903 | 2018.07.09 08:10 | 90.9 | 1243.8 | 711.7 |
| 2018070904 | 2018.07.09 08:29 | 89.5 | 1235.5 | 691.6 |
| 2018070905 | 2018.07.09 09:34 | 93.1 | 1752 | 1105.5 |
| 2018071201 | 2018.07.12 04:16 | 91.4 | 1247.1 | 701.3 |
| 2018071202 | 2018.07.12 04:31 | 94.8 | 1246.1 | 721.7 |
| 2018071203 | 2018.07.12 07:09 | 92.7 | 1246.5 | 719.6 |
| 2018071204 | 2018.07.12 07:33 | 93.2 | 1240.9 | 717.8 |
| 2018071205 | 2018.07.12 09:44 | 98.6 | 1253.7 | 722.3 |
| 2018071206 | 2018.07.12 14:30 | 92.8 | 1248.9 | 721.3 |
| 2018071207 | 2018.07.12 16:30 | 92.9 | 1240.2 | 716.5 |

Besides, a radiosonde balloon which was operated by the Institute of Energy and Climate Research - Stratosphere (IEK-7) mea-
sured the atmospheric parameters from ground to 25 km altitude. COBALD was part of a CFH / ECC ozone / RS41 payload to
provide the backscatter coefficients as well as air temperature, air pressure, relative humidity, and wind with the temporal and spatial resolution being 1s and around 5 m vertically.
The COBALD is a lightweight (500 g) aerosol backscatter detector for balloon-borne measurements developed at the Insti-
tute for Atmospheric and Climate Science (ETH Zürich), based on the original approach by Rosen and Kjome (1991). Two light-emitting diodes (LEDs) as light sources and a photodiode detector with a FOV of 6° provide high-precision *in-situ*



measurements of aerosol backscatter at wavelengths of 455 nm (blue visible) and 940 nm (infrared). COBALD has been orig-
inally developed for the observation of high-altitude clouds, such as cirrus (Brabec et al., 2012; Cirisan et al., 2014) and polar
stratospheric clouds (Engel et al., 2014), while recently it was proven able to detect and characterize aerosol layers in the
upper troposphere–lower stratosphere (Vernier et al., 2015, 2018; Brunamonti et al., 2018, 2021). In this work, we compared
COBALD measurements with scanning aerosol LIDAR measurements for validating LIDAR retrievals and investigating the
vertical distribution of aerosols. A summary of sensors used on UAV and balloon fights is shown in Table 2.

**Table 2.** Summary of sensors used on on UAV and balloon fights.

| | Measurement | Instrument | Manufacturer | sample flow (lpm) | Time resolution | Mode of operation |
|---|---|---|---|---|---|---|
| **UAV** | | | | | | |
| | Particle size distribution (0.35 - 40 $\mu$ m) | OPC-N3 | Alphasensse | 5.5 | 1.6 s | 24 size bin |
| | T, RH | ChipCap2 sensore | Telaire | | 1s | |
| | Pressure, wind speed & direction | eBee sensors | AgEagle Aerial Systems Inc. | | 1 s | |
| | Lat, lon, | | | | 1 s | |
| **Balloon** | | | | | | |
| | Backscatter ratio (455 nm & 940 nm) | COBALD | IAC (ETH, Zürich) | | 1 s | |
| | Ozone | Electrochemical concentration cell (ECC) | JOSIE (Smit et al., 2007) | | 1s | |
| | Water vapor | Cryogenic frostpoint hygrometer (CFH) | EnSci (Vömel et al., 2007; Vömel et al., 2016) | | 1s | |
| | Temperature | Vaisala RS41-SGP | Vaisala | | 1s | |
| | Altitude, lat, lon & horizontal wind | | | | 1s | |






# 3 Results and Discussion

## 3.1 Comparison of LIDAR data with ground level *in-situ* measurements

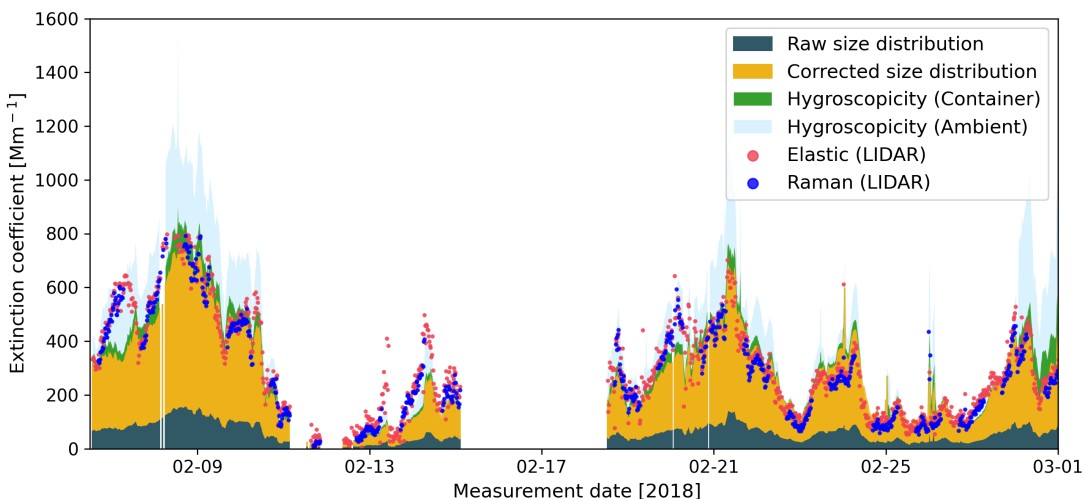

**Figure 2.** Time series of ground-level extinction coefficients retrieved from LIDAR measurements (both elastic and Raman methods), Mie calculation based on OPC raw size distribution as well as size distribution corrected by counting efficiency and hygroscopic effect from February $5^{th}$ to March $5^{th}$, 2018 in downtown Stuttgart.

The comparison of LIDAR retrievals with ground-level aerosol sizer data was conducted during a field campaign from February $5^{th}$ to March $5^{th}$, 2018 in downtown Stuttgart. In this campaign, the aerosol LIDAR did zenith scans with an ele-
vation angle from 90 ° to 5° in steps of 5°. The nearly horizontal measurement at 5° allows to retrieve extinction coefficients near ground level (from 25 m to 50 m above ground level) by using short-range LIDAR data (ranges: 285 m to 570 m) that can be compared with the ground-level in situ measurements (sampled 3.7 m above ground level). The ground-level *in-situ* aerosol sizer, Fidas200, measured the aerosol size distributions which were used to calculate the aerosol extinction coefficients via Mie code. Figure 2 shows the extinction coefficients retrieved from LIDAR measurements and from Mie calculations based
on aerosol size distribution (labeled as "Raw size distribution"). The extinction coefficients obtained from LIDAR were both retrieved from the slope and Raman retrieval methods (Seidel et al., 2010; Ansmann et al., 1992). In the slope and Raman retrieval methods, a linear regression was used and the correlation coefficients of linear regressions are $0.99 \pm 0.05$ and $0.99 \pm 0.06$ for slope and Raman retrieval methods, respectively. This is also an indication for a rather homogeneous distribution of the aerosol particles within the altitude range from 25 to 50 m corresponding to a range between 285 and 570 m. This figure shows
that the raw extinction coefficients from Mie calculations are systematically lower than those from LIDAR retrievals by a factor of $4.70 \pm 1.49$. The reason for this phenomenon is that the Fidas200 underestimates the particle number by a factor of 2-10 at a diameter between 0.25 $\mu$m and 0.5 $\mu$m when compared with SMPS data as shown in Figure S1. The left side of Figure S1

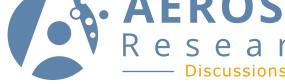

shows the number size distribution from Fidas200 and the merged size distribution from SMPS and APS measurements. From this figure, we can see that Fidas200 underestimated particle number size distributions at a diameter between 0.25 $\mu$m and 0.5

$\mu$m when compared with the merged size distribution (called "loss effect"). The right plot of this figure shows the accumulated extinction coefficients calculated from Mie based on those two size distributions, which shows that the underestimation of particle numbers from 0.25 $\mu$m to 0.5 $\mu$m causes the modelled extinction from the Fidas200 size distributions to be lower than that modelled from merged size distribution by a factor of around 4. Hence, we conclude that the underestimation of particle number from 0.25 $\mu$m to 0.5 $\mu$m is one of the main reasons for the underestimation of extinction coefficients based on OPC

data alone.

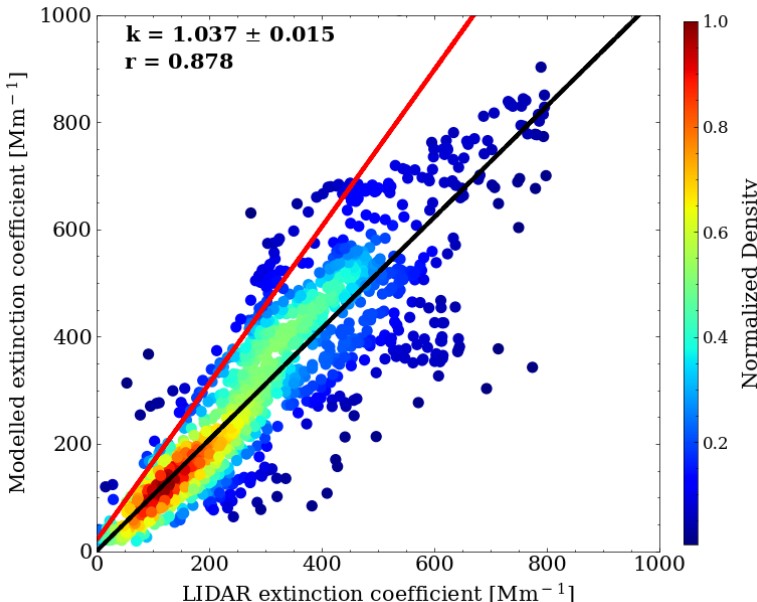

**Figure 3.** Correlation of extinction coefficients from LIDAR retrieval and Mie calculation from February $5^{th}$ to March $5^{th}$, 2018 in Stuttgart. The relative humidity used in the model is container indoor relative humidity and the black line is the regression fitting curve of them. The red line is the regression fitting curve between the LIDAR-derived extinction coefficients and those from Mie calculation by using ambient relative humidity.

The systematic underestimation of aerosol particle number from 0.25 $\mu$m to 0.5 $\mu$m allows for calculating a counting efficiency curve as shown in Figure S2. Then the calculated counting efficiency curve was applied to the Fidas200 size distribution to get a corrected aerosol size distribution. This corrected size distribution is used to calculate the corrected extinction coefficients via

Mie calculation. The time series of the corrected extinction coefficients calculated from the corrected size distribution is shown in Figure 2. The orange area indicates the extinction coefficients due to the underestimation of aerosol particle number from 0.25 $\mu$m to 0.5 $\mu$m. After taking into account the particle number underestimation, the modelled extinction coefficient shows





good agreement with LIDAR retrievals. Although good agreement between *in-situ* and LIDAR measurements, the aerosol hygroscopic growth effect is still not considered. The modelled extinction coefficients contributed by aerosol hygroscopic growth

are labeled as "hygroscopicity (container)" and "hygroscopicity (Ambient)" in Figure 2, representing the relative humidity used in the model are container indoor relative humidity and ambient relative humidity, respectively. The correlation plot between the extinction coefficient for container indoor relative humidity and the LIDAR-derived extinction coefficient is shown in Figure 3, which shows a slope and a Pearson correlation coefficient of $1.037 \pm 0.015$ and $0.878$, respectively. The dashed line in this figure is the regression fitting curve between the LIDAR-derived extinction coefficients and those from Mie calculation by

using ambient relative humidity, which shows a slope and a Pearson correlation coefficient of $1.463 \pm 0.025$ and $0.845$, respectively. As shown in these two figures, the extinction coefficients retrieved from LIDAR measurement show a similar trend for both extinction coefficients but shows better agreement with the one calculated based on the container indoor relative humidity. The reason for a better agreement based on the indoor relative humidity instead of the outdoor ambient relative humidity is due to the fact that the aerosol particles lost their water partly inside the container but did not reach equilibrium within the 3

s residence time in the sampling line. Please note, that there was no dryer in the sampling line and the impact of the relative humidity correction on our comparison is much smaller than the correction of the size measurements. From the fraction of extinction coefficients shown in Figure 2, we can determine that the main reason for causing extinction coefficient inconsistency between *in-situ* measurement and LIDAR retrieval is the undercounting by the OPC. The relatively good agreement of the extinction coefficients after our reasonable corrections reflects the reliability of our methods and the good quality of the

LIDAR retrievals.

### 3.2  Comparison of LIDAR data with *in-situ* measurements on a UAV

The comparison of LIDAR and UAV measurements was conducted for two days, on July $9^{th}$ and July $12^{th}$, 2018 to study the vertical distribution of aerosols and the boundary layer structure. The sky was almost free of clouds during UAV flights on July $9^{th}$ while it was affected by clouds within the boundary layer on July $12^{nd}$. Figure 4 shows the time series of backscatter

coefficients and boundary layer retrieved from LIDAR measurement (pink squares) as well as boundary layer height (a.s.l. - above see level) obtained from ERA5 dataset (white dashed line) and potential temperatures obtained from UAV measurements (white solid line) on July $9^{th}$, 2018. This figure shows that the boundary layer height retrieved from the LIDAR measurement is consistent with the boundary layer height from the UAV measurement (the maximum gradient of potential temperature) which both show an increasing trend of the boundary layer during the morning of this day. In addition, the boundary layer from ERA5

also shows a similar trend as the observations but overestimates boundary layer height, especially during daytime. A possible reason for this overestimation is that the existence of clouds during daytime reduced solar radiation and a low value of solar radiation caused a shallow boundary layer at this time. Figure 4 also shows a stable nocturnal boundary layer and a residual layer during nighttime measured by scanning aerosol LIDAR. The low and stable boundary at night time can suppress the dispersion of aerosol near the surface. Hence, the backscatter coefficients within the boundary layer are maximum (highest aerosol

concentration) during the morning rush hour due to the combined effect of the shallow boundary layer and local anthropogenic emissions. After sunrise, the convection became stronger, which caused an increase of the boundary layer height and dilution of



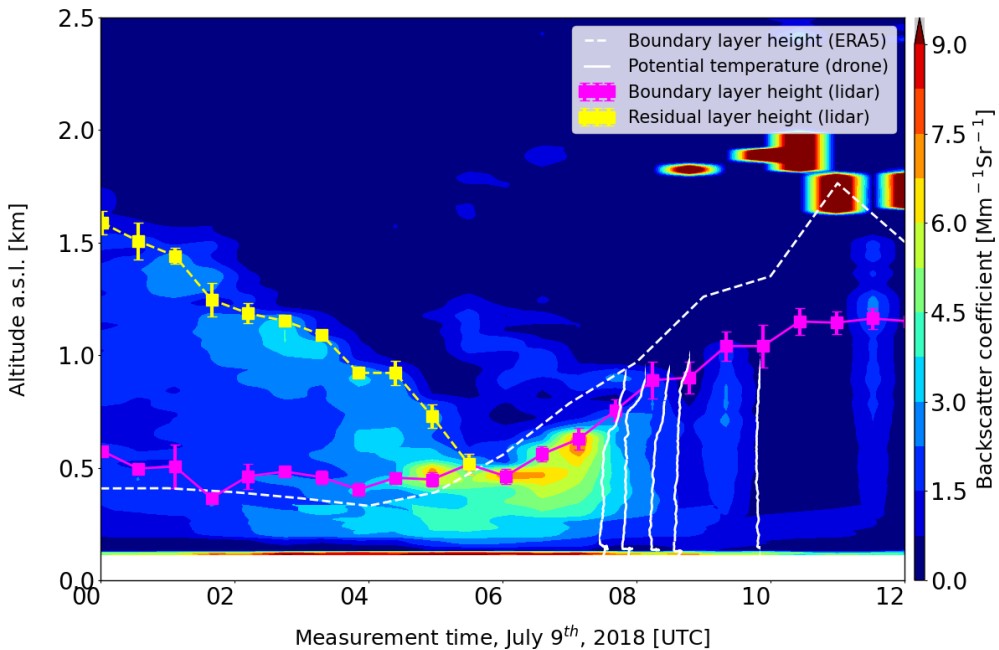

**Figure 4.** Time series of backscatter coefficients (contour), boundary layer height (pink squares) and residual layer retrieved (yellow squares) from scanning LIDAR, as well as boundary layer heights obtained from ERA5 dataset (white dashed line) and vertical potential temperature profiles (white solid line) measured by UAV on July $9^{th}$, 2018.

aerosols within the boundary layer, so the aerosol concentrations within the boundary layer decreased. Figure 5 shows the time series of range-corrected LIDAR signal and boundary layer heights retrieved from LIDAR as well as boundary layer height obtained from ERA5 dataset (white dash line) and potential temperature obtained from UAV measurements (white solid line)

on July$12^{th}$, 2018. The reason for showing range-corrected LIDAR signal instead of backscatter coefficients is that low-level clouds prevented retrieving the backscatter coefficients from range-corrected LIDAR signal by the Klett-Fernald method. This figure also shows consistency in boundary layer heights among LIDAR, UAV, and ERA5. More interestingly, the cloud existed at the top of the boundary layer from 05:00 to 13:00 and the cloud base increased with boundary layer height as captured by the LIDAR measurements. The reason for the cloud existing on the top of the boundary layer is that the relative humidity has

a maximum value at the top of the boundary layer in the well-mixing boundary layer and this high relative humidity ambient environment provided a good conditions for cloud formation. Figure S3 shows the correlation of boundary layer heights from LIDAR and radiosonde retrievals for both two days, which show a good correlation with a slope of $1.01 \pm 0.24$ and a Pearson correlation coefficient of 0.793.

A comparison of the vertical profile of aerosols from LIDAR and UAV measurements was conducted in the following steps. First, we used the temperature and pressure measured by UAV instead of an atmospheric model to calculate molecular backscat-





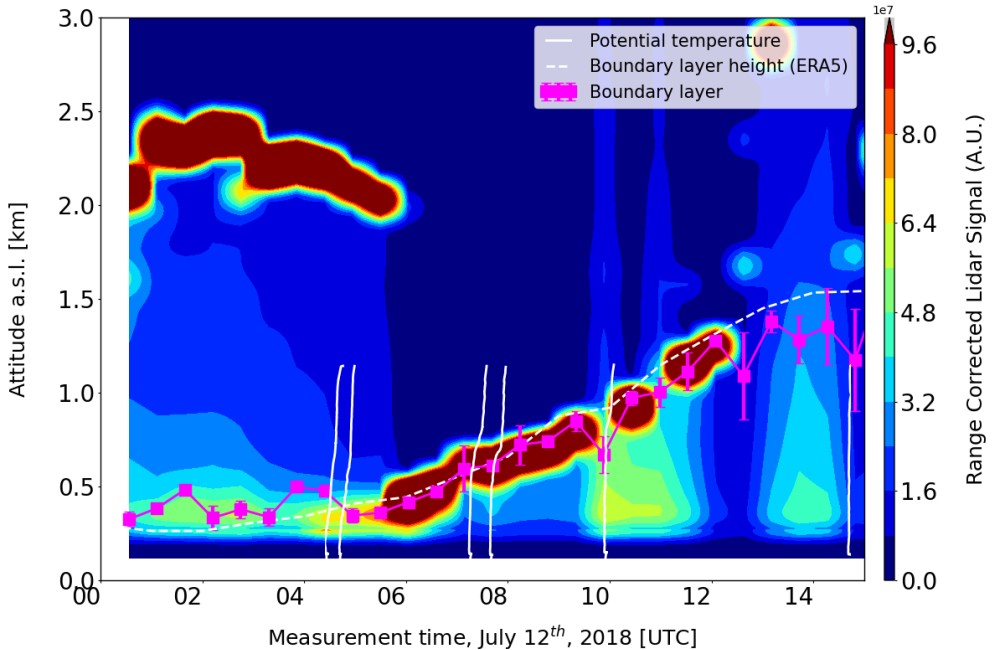

**Figure 5.** Time series of range corrected LIDAR signal and boundary layer height retrieved from scanning LIDAR (pink squares) as well as boundary layer heights obtained from ERA5 dataset (white dashed line) and vertical potential temperature profiles (white solid line) measured by UAV on July 12$^{th}$, 2018.

ter coefficients, and these molecular backscatter profiles were used for LIDAR retrievals. Second, the backscatter coefficients at all observation angles were calculated using the Klett-Fernald method with reference values obtained from vertical profiles of the backscatter coefficients. Finally, Mie theory was used to calculate the aerosol backscatter coefficients based on size

distributions measured by the UAV-borne OPC and the complex refractive index from a nearby sun photometer. As there are no dryer before OPC-N3 sampling and no temperature difference between sampling tube and ambient environment, the effect of relative humidity on aerosol sampling was not considered. Figure 6 shows the backscatter coefficients retrieved from LIDAR measurements and from Mie calculations based on size distributions measured by the OPC on the UAV. In this experiment, the LIDAR performed zenith scans using elevation angles from 90° to 5° with steps of 5° during the UAV flights. Consequently,

we retrieve the backscatter coefficients for each observation angle and the average of these backscatter coefficients is shown as thick red line to compare with the UAV measurements. This figure shows that the vertical distribution of the aerosol particles in the well-mixed boundary layer is reflected well in both LIDAR and OPC measurements. Furthermore, the backscatter coefficients from UAV retrievals (green dashed line in figure 6) show the same aerosol mixing height and the same order of backscatter coefficients as LIDAR retrievals. The smaller backscatter coefficients calculated based on airborne OPC measure-

ments may be related to an undercounting of the smaller particles as we have seen for ground based OPC measurements by the Fidas 200 instrument. The size distributions were corrected (black dashed line in figure 6) by the counting efficiency curve

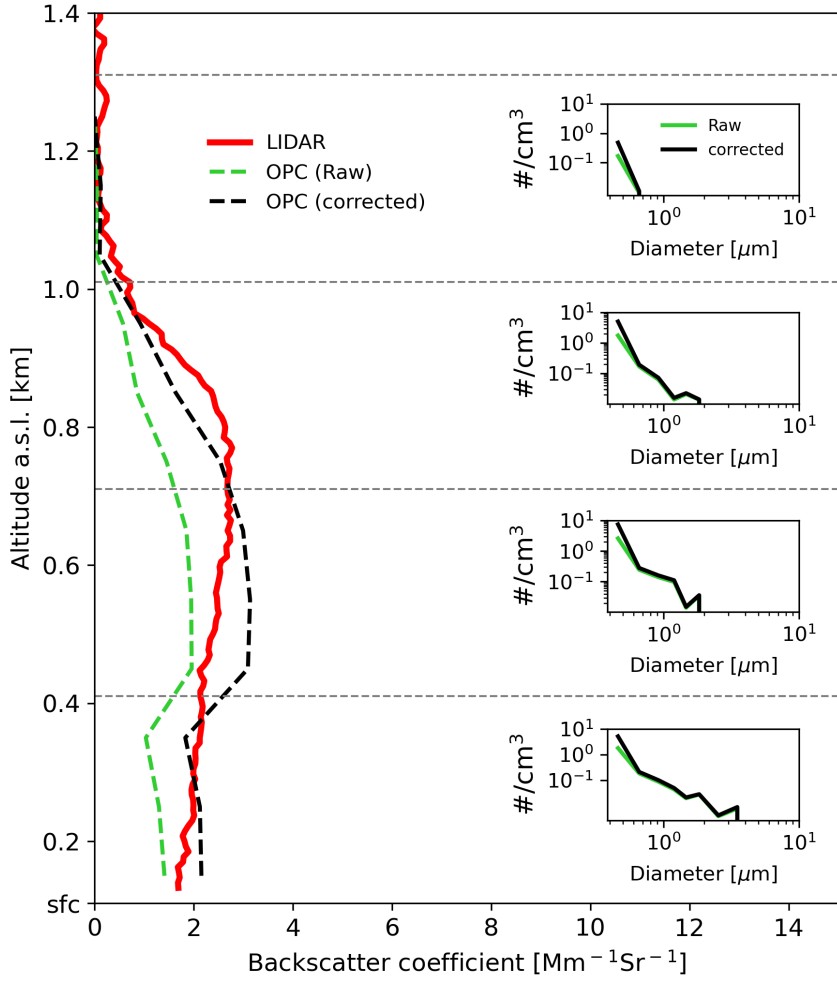

**Figure 6.** Vertical distribution of backscatter coefficients from LIDAR measurement (solid red line), as well as backscatter coefficients derived from UAV-based measurements for raw size distributions (dashed green line), and corrected particle size distributions (dashed black line) (inserts on the right) on July $9^{th}$, 2018. Note: The 'sfc' on the y-axis indicates ground surface level.

introduced in section 3.1. The backscatter coefficients from corrected size distributions were consistent with the lidar-derived backscatter coefficients. Although Fidas200 is a different OPC sensor as OPC-N3, the same undercounting phenomenon was observed for both sensors. Please note that the particle size is averaged over 300 m and the horizontal dashed lines represent these average altitude ranges. These vertical size distributions show that larger particles were detected only below 300 m above ground level.

12 UAV flights were conducted on July $9^{th}$ and July $12^{th}$ as shown in Table 1 to compare with LIDAR retrievals. Figure



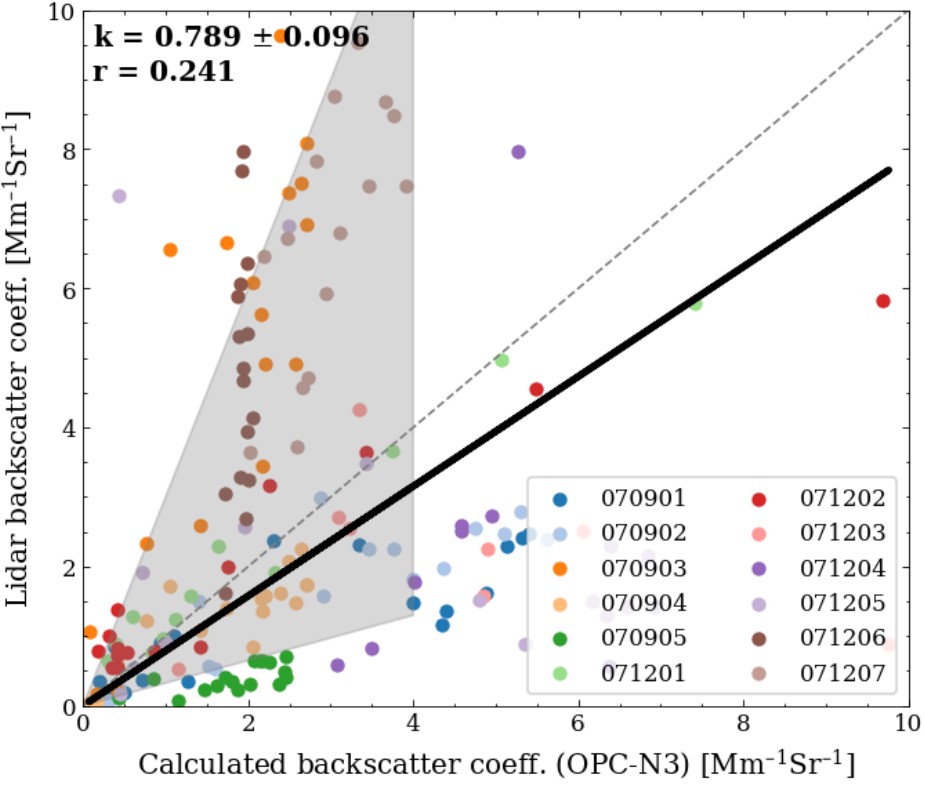

**Figure 7.** Correlation of backscatter coefficients retrieved from LIDAR measurement and modelled from Mie calculation based on aerosol size distribution measured by OPC-N3 on the UAV for all UAV flights on July $9^{th}$ and July $12^{th}$, 2018. The different scatter point colours indicates different UAV flights. The thick black line is a linear fit to the data and the thin dashed line is the 1:1 line

7 shows the correlation of backscatter coefficients retrieved from LIDAR measurement and from Mie calculation based on
aerosol size distributions measured by OPC-N3 on the UAV. The data from LIDAR and UAV was averaged into 60 m vertical
bins to reduce the noise of the OPC-N3 measurement. The colours of the scatter points indicated different UAV flights. This
figure shows that the backscatter coefficients retrieved from LIDAR correlated on average with the backscatter coefficients
calculated from the OPC with a slope of $0.789 \pm 0.096$ and a Pearson correlation coefficient of 0.234. This figure also shows
that 75% of data points are within the grey shaded area, which indicates that these data are within a factor of 3. However, in
contrast to the ground level OPC measurements a dedicated correction of the low cost OPC data for potential sampling artefacts
or undercounting was not possible. This figure also shows that the UAV measurements reflect the same aerosol mixing process
within the boundary layer and the same order of magnitude of the backscatter coefficient. However, the backscatter coefficients
retrieved from UAV-borne OPC in certain UAV flights still show a relatively large deviation from LIDAR retrievals in certain
flights. One reason for these unstable observations is that the UAV cruising speed may affect aerosol sampling by the OPC-N3.

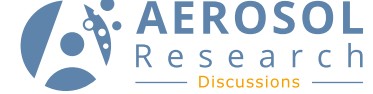

The sample was collected perpendicular to the flight's direction into the OPC, so we can expect size-dependent discrimination of larger particles.

### 3.3    Comparison of LIDAR data with *in-situ* measurements onboard a balloon

A balloon which carried the COBALD sensor to measure backscatter coefficients *in-situ* was launched to an altitude of around 30 km on the night of July $12^{th}$, 2018 to validate LIDAR retrievals. The LIDAR did vertically pointed measurements with
an integration time of 60 s for each profile during the balloon launch. Figure 8a shows the range corrected LIDAR signal for two hours of continuous measurement and the vertical trajectory of the balloon. As shown in this figure, the LIDAR signal did not vary much in the first hour (the period was highlighted in this figure) while showing changes in the second half of the experiment. Hence, we selected the first hour to compare with balloon measurements. Figure 8b shows the horizontal trajectory of the radiosonde with the colour of the plot indicating the radiosonde altitude and the circle indicating the distance from the
LIDAR observation station. This figure shows that the horizontal displacement of the radiosonde is about 10 km when the radiosonde reached an altitude of 10 km and this horizontal displacement may cause a difference in backscatter coefficients between LIDAR and COBALD. For the LIDAR analysis in this experiment, the backscatter coefficients were retrieved from elastic and Raman data with the vertical profiles of the molecular backscatter coefficient being calculated from temperature and pressure measured by the balloon. The COBALD data analysis follows the procedure proposed by Brunamonti et al. (2021).
First, a wavelength extrapolation yielded the backscatter coefficient at a wavelength of 355 nm from COBALD measurement. The Ångström exponent (AE) used for this wavelength conversion is measured by COBALD at two wavelengths (455 nm & 940 nm) and extended to the wavelength of 355 nm. Second, as the Field of View (FOV) of LIDAR and COBALD are different (the FOV of COBALD is 6° whereas the FOV of LIDAR is 2.3 mrad), a FOV correction is necessary. The correction factors are calculated based on Mie theory and are shown in Figure 2 in Brunamonti et al. (2021).
Figure 9 shows the backscatter coefficients from COBALD and LIDAR measurement for a LIDAR integration time of 1 hour. These two profiles of backscatter coefficients from LIDAR are retrieved from elastic and Raman channel data respectively.

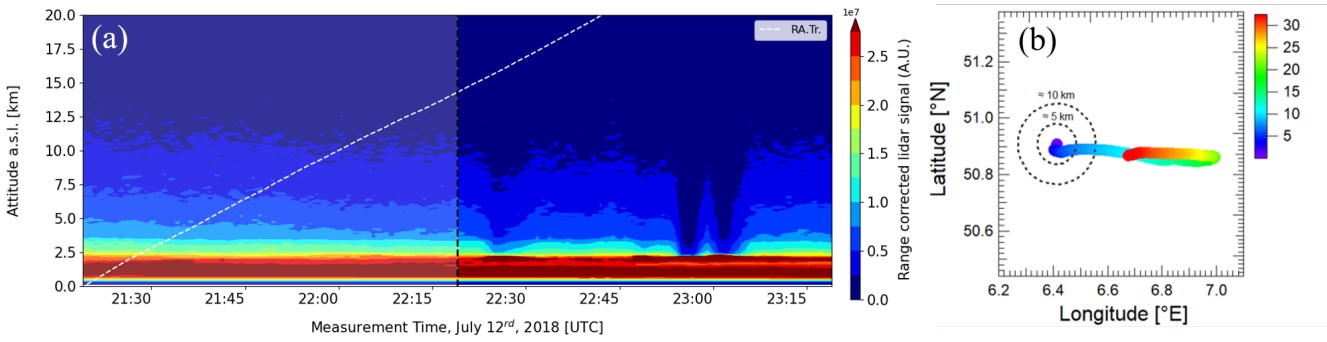

**Figure 8.** Time series of range corrected LIDAR signal and radiosonde vertical trajectory (white dash line) (a) and Horizontal displacement of the balloon during this experiment (b) on July $12^{th}$, 2018 at Jülich research center.

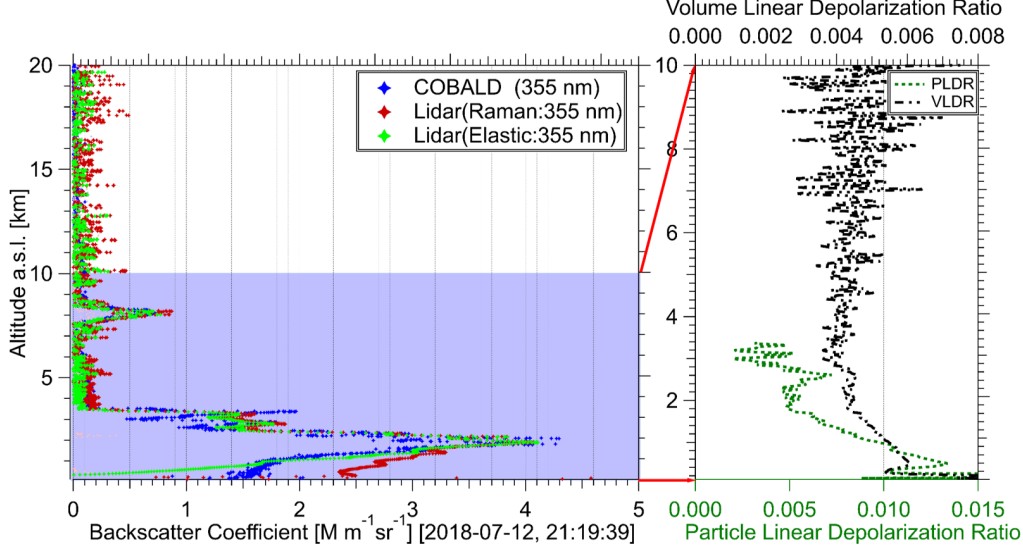

**Figure 9.** Backscatter coefficients measured by balloon-borne COBALD and LIDAR (left) as well as aerosol volume and particle depolarization ratio measured by LIDAR (right) on the night time of July $12^{th}$, 2018 at Jülich research center. (The integration time of the LIDAR data is 1 hour from 21:19 to 22:19.)

The retrieval of backscatter coefficients from elastic channel data remained with larger uncertainty due to the assumption of a LIDAR ratio in the Klett-Fernald method. Hence, it is more meaningful to compare backscatter coefficients from Raman data with those from COBALD measurements. In addition, the volume and particle depolarization ratios measured by LIDAR

are shown on the right side of Figure 9. The low depolarization ratios support our assumption that the particles are spherical and that we can use Mie calculations for the FOV correction. This figure shows a good agreement in backscatter coefficients between LIDAR Raman data retrieval and COBALD measurement at an altitude above 2 km. However, there is a significant discrepancy at altitudes below 2 km.

The discrepancy of the backscatter coefficients between LIDAR retrievals and COBALD measurements at lower altitudes is

due to the temporal evolution of aerosol particle concentrations in the boundary layer as can be seen from vertical profiles of backscatter coefficients with high temporal resolution in Figure S4. This figure shows profiles of backscatter coefficients retrieved from LIDAR Raman data with 5 - minute temporal resolution and backscatter coefficients measured by COBALD as well as the vertical balloon trajectory. This figure shows a good agreement in backscatter coefficients between COBALD measurement and LIDAR Raman data retrievals at the altitude of the balloon passing by. The backscatter values at the altitude

of the balloon passing by are extracted as shown as the red line in Figure S4 to obtain merged Raman backscatter coefficients. The merged Raman backscatter coefficients and backscatter coefficients from COBALD measurements are shown on the left side of Figure 10, showing very good agreement of backscatter coefficients from LIDAR and COBALD measurements at all altitudes. The correlation between LIDAR merged Raman backscatter coefficients and COBALD backscatter coefficients is




shown on the right side of Figure 10, which shows these two backscatter coefficients are well correlated with a slope of 1.063

± 0.016 and a Pearson correlation coefficient of 0.925. This consistency between LIDAR and COBALD sensor reflects a good data quality of both methods and proves that LIDAR can provide reliable and vertical profiles of aerosol particles with high spatial-temporal resolution.

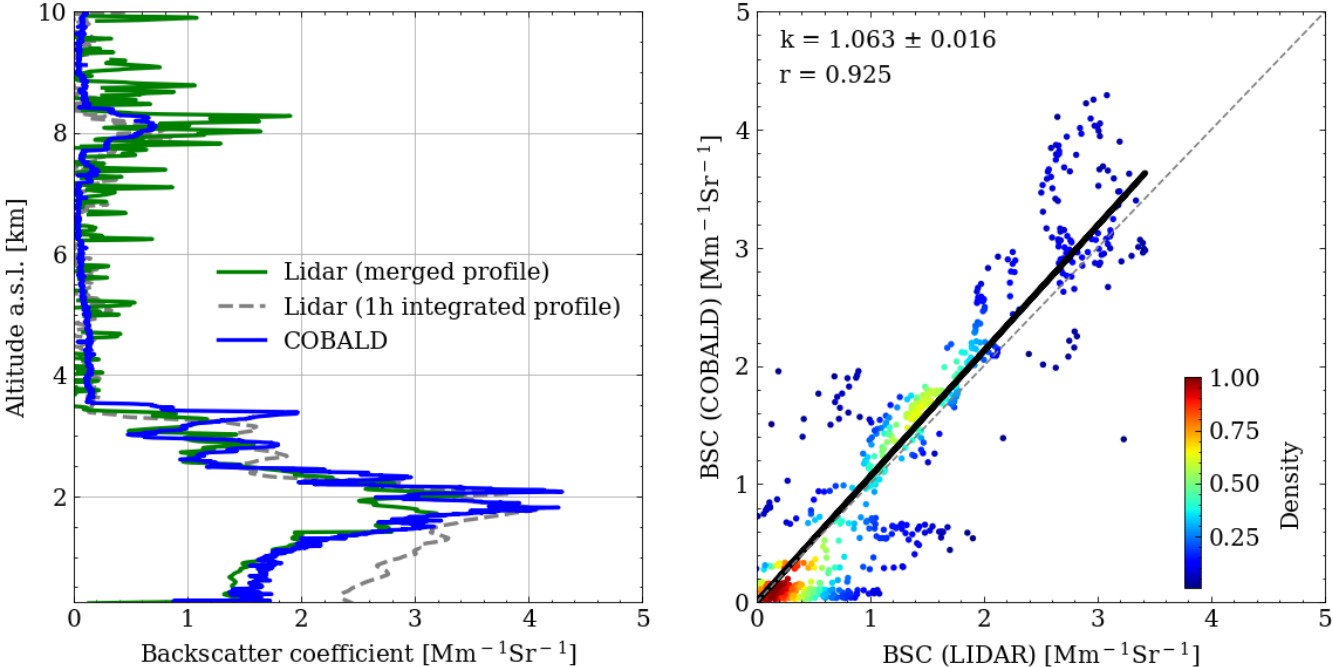

**Figure 10.** Profiles of backscatter coefficients from LIDAR for integration over 1 hour (grey dash line) and sliding 5-minute merged backscatter coefficients (green line) as well as the vertical profile of *in-situ* backscatter coefficient measured by balloon-borne COBALD (blue line) on July 12$^{th}$, 2018 at the Jülich research center (left). Correlation between LIDAR merged backscatter coefficients and balloon-borne COBALD backscatter coefficients (right).

## 4   Conclusions

This paper presents results of aerosol spatial-temporal distribution and optical properties measured by a scanning aerosol LIDAR, a radiosonde with a backscatter sensor, an OPC-N3 on a UAV, and a comprehensive set of ground level *in-situ* measurements. Modern aerosol characterisation methods including remote sensing and *in-situ* methods helped us better understand the aerosol physical properties and build a bridge between remote sensing and these *in-situ* methods. This paper focuses on the comparison of aerosol measurement between LIDAR retrievals and *in-situ* measurements at ground level, in the troposphere,

and in the stratosphere, thus validating LIDAR retrievals at all altitude levels.

The comparison of ground-level *in-situ* extinction coefficients with LIDAR-derived ones shows that Fidas200 underestimated particle number concentration by a factor of 2-10 at the diameter range between 0.25 $\mu$m and 0.5 $\mu$m, thus causing the total extinction calculated from this size distribution to be systematically lower than that from LIDAR retrievals by a factor of 4.70 $\pm$ 1.49. The extinction coefficient calculated from the Fidas200 aerosol size distribution corrected by SMPS size distribution

shows good agreement with LIDAR-derived extinction coefficient with a slope of 1.037 $\pm$ 0.015 and a Pearson correction coefficient of 0.878. The comparison also shows that the undercounting of aerosol particles between 0.25 $\mu$m and 0.5 $\mu$m is the main reason for the large discrepancy between LIDAR retrieval and ground-level *in-situ* Fidas200 measurements. In addition, a comparison between LIDAR and UAV shows good agreement in boundary layer height measurements and both methods show a similar trend as the ERA5 boundary layer height evolution. The OPC-N3 aboard UAV shows a similar aerosol vertical

distribution and comparable backscatter coefficients as LIDAR measurement. However, the backscatter coefficients calculated from OPC-N3 were unstable and large uncertainties still remained for different flights most likely due to the effect of UAV cruising on OPC-N3 sampling. Adapting the inlet design of the OPC may improve the data quality for future measurements. Finally, the backscatter from balloon-borne COBALD measurement shows very good agreement with the backscatter retrieved from LIDAR measurement if compared with 5-minute resolution LIDAR data with a slope of 1.063 $\pm$ 0.016 and a Pearson

correlation coefficient of 0.925. This consistency between LIDAR and COBALD sensor validated our LIDAR retrievals and proves that LIDAR can provide reliable and high-resolution vertical profiles of aerosols. In conclusion, the retrievals from scanning aerosol LIDAR measurements show good agreement with *in-situ* measurements at all altitude levels and these LIDAR measurements can also used as reference for other low cost in-situ measurements.

*Code availability.*  The code used to analyse the LIDAR data is property of Raymetrics Inc, but we have shown that it gives the same results

as the code "single calculus chain" (SCC) provided by EARLIENT https://www.earlinet.org/index.php?id=earlinet_homepage, last access: 14 February 2023 and public available. The Mie code used in this paper is available via github repository https://github.com/jleinonen/pymiecoated, last access: 14 February 2023.

*Data availability.*  The LIDAR raw data and ground *in-situ* data are available via the open access data repository KITopen (link to be added). The UAV data and balloon data are available via the open access data repository Jülich DATA (link to be added).

*Author contributions.*  CR, RT, CW, and HS performed the measurements and analyzed *in-situ* measurement data. HZ analysed the LIDAR remote sensing data. FGW post-processed the COBALD data. HZ wrote the manuscript with support from HS as well as contributions from all co-authors.

*Competing interests.*  The authors declare that they have no conflict of interest.

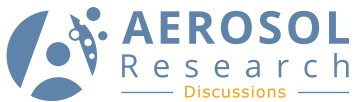

*Acknowledgements.* Support by the staff of the Institute of Meteorology and Climate Research and the Institute of Energy and Climate
Research (FZJ), financial support by the project Modular Observation Solutions for Earth Systems (MOSES) of the Helmholtz Association
(HGF).





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
