# Peer review of "Comparison of scanning aerosol LIDAR and *in-situ* measurements of aerosol physical properties and boundary layer heights"

_Aerosol Research, 2024_

## Author Response (AR1)

**Reply to the referee's comments**

We thank the referee for the useful comments, which helped us to improve the quality of our manuscript.

In the following, the referees' comments are given in black.

Our point-to-point replies are marked by "R" and are in blue.

Changes to the manuscript text are in green.

**Comments by referee #1**

The focus of the manuscript is on the comparison of (a) almost horizontal aerosol lidar observations with respective in situ aerosol observations at ground, of (b) lidar-derived optical properties with respective UAV OPC measurements, and (c) of lidar profiles of aerosol backscatter up to the tropopause with respective balloonborne COBALD backscatter sonde observations.

The experimental effort is large and that aspect deserves publication. However, the results are not new. Lidar comparisons with COBALD observations and with in situ aerosol observations have already been performed and published. The use of a scanning lidar in these comparisons may be a new aspect.

The manuscript provides the impression that the goal is to validate lidar observations. However, it is known at least since the 1980s that lidar observations are trustworthy. To my opinion, the lidar data (presented here) confirm that in situ aerosol observations can be used (after proper corrections of instrumental effects and other short comings) to compute optical properties. Furthermore, the lidar observations show that the COBALD sonde allows proper aerosol measurements. But the high quality of COBALD observations is also known since many years.

R: We agree that several comparison studies with lidar measurements have been conducted previously. However, this is not the case for the scanning Raman lidar that we used in this study. Given the special capabilities of the scanning lidar we think it is valuable for the readers to learn how it compares to other methods.

1. The scanning lidar allows to conduct horizontal measurements which facilitates direct comparison with ground level observations (Shin et al. 2024). This direct comparison with ground level observations allows us to do long-term comparisons with high time resolution (e.g 10-minute resolution over one month), which facilitates a robust statistical evaluation of this kind of comparison. Furthermore, employing multi-angle analysis of an aerosol layer allows to retrieve backscatter coefficients independently without needing to assume a lidar ratio (Zhang et al. 2021). Since the scanning lidar retrieval algorithm isn't very common we think it is justified to demonstrate its quality experimentally.

2. As far as we know, this manuscript firstly evaluates the comparison between scanning lidar and *in-situ* measurements over a wide altitude range from ground level to the stratosphere (around 20 km). This comparison over a wide altitude range includes the high variability of aerosol concentrations within the boundary layer and relatively stable concentrations in free troposphere, which shows the reader the relevant time-scales and spatial-scales of aerosol evolution when conducting such comparison.

3. Compared with previous literature, this manuscript not only focuses on the comparison of remote sensing and *in-situ* measurements but also discusses the potential meteorological factors that affect the comparison results (section 3.3). We have modified the end on the introduction as follows:

However, to our best knowledge, so far no dedicated comparison of scanning LIDAR measurement with *in-situ* observation has been performed over a wide altitude range and over such a long time period for comparison at ground level (e.g. one month dataset with 10 minute resolution). Also in order to bridge the gaps that are often encountered between remote sensing and *in-situ* observation, we compared datasets on aerosol spatial-temporal distributions and evolution combining remote sensing and *in-situ* measurements.

**Detailed comments:**

1. p2, l25: you mean ... climate change simulations...

R: Thank you for pointing out this. we have changed "climate change" to "climate change simulations".

these uncertainties have a great impact on climate change simulations.

2. p2, p2, l49: CALIPSO is a simple backscatter lidar. Advanced techniques are the HSRL and Raman lidar techniques. That should be mentioned.

R: We have added the recent status of two new satellite missions : ACDL (China) and EarthCare (ESA and JAXA). Both satellite missions deploy HSRL. Furthermore, we added one new paragraph to introduce different advanced lidar techniques.

China launched its first space-borne aerosol-cloud high-spectral-resolution lidar (ACHSRL) on April 16, 2022, which is capable for high accuracy profiling of aerosols and clouds around the globe (Ke et al. 2022). Also, the Earth Cloud, Aerosol and Radiation Explorer (EarthCARE) is a satellite mission implemented by the European Space Agency (ESA), in cooperation with the Japan Aerospace Exploration Agency (JAXA), to measure global profiles of aerosols, clouds and precipitation properties together with radiative fluxes and derived heating rates, due for launch in May 2024 (Wehr et al. 2023).
Aerosol elastic scattering lidar is widely used in lidar observation networks as it can provide detailed information with high spatial and temporal resolution. However, retrieving backscattering coefficients from this kind of lidar data requires assumptions of lidar ratios and reference values (Fernald 1984, Klett 1985). One of the successfully used technology to overcome this problem is the Raman lidar (Wandinger 2005, Groß et al. 2015, Baars et al. 2016, Hu et al. 2022). Another widely used technology is the high spectral resolution lidar (HSRL) (Liu et al. 1999) which used narrow-band filter (e.g. atom or molecule filter) to separate signals from molecule and particle backscatter (Piironen & Eloranta 1994). And this HSRL allows us better to investigate aerosol optical properties (Burton et al. 2012, 2014, Groß et al. 2013). Recently, a HSRL that uses an interferometer as filter has been deployed at other wavelengths. The recently launched Doppler Wind Lidar, ALADIN, uses this technology to measure tropospheric wind profiles on a global scale but can also obtain vertical aerosol profiles (Schillinger et al. 2003).

3. p2, l41: Kotthaus et al. does not focus on aerosol optical properties, only on PBL heights.

R: We have changed corresponding references.

In addition to these satellite missions, ground-based remote sensing methods are used to investigate aerosol optical properties (Adam et al. 2020, Mylonaki et al. 2021).

4. p2, l45: MPLNET consists of simple backscatter lidars. . .

R: Although MPLNET consists of simple backscatter lidars, it has quite some coverage and should be mentioned. In addition, we added a new paragraph on state-of-the-art HSRL and Raman lidars.

5. Nothing is mentioned regarding state-of-the-art HSRL and Raman lidars and their potential (Burton et al., 2012, 2015, Gross et al., 2011, 2015, Baars et al. 2016, Hu et al. 2022) and many other papers, e.g., from Veselovskii and others. . .

R: Thank you for pointing to these publications. We have added them in the new paragraph on these state-of-the-art HSRL and Raman lidars as given in our answer to your comment 2 above.

6. p3, l62: Ceolato and Berg, 2012 review article: This is quite a poor review article, mainly covering just the simplest of all aerosol lidar techniques.

R: We omit this reference.

7. p3, l62: p3, l63: 2 times Duesing et al., 2018, is mentioned.

R: We correct the doublication.

8. All in all: The introduction does not reflect the latest status of lidar efforts, studies, and publications in the field of aerosol remote sensing.

R: We have added a new paragraph on the recent development of these state-of-the-art HSRL and Raman lidars including related publications as given in our answer to your comment 2. Furthermore, we added four more references on previous comparisons of in-situ and lidar measurements to the introduction.

9. p5, l131: The boundary layer detection method (Haar wavelet transform method) has often been presented. Please omit, just provide a reference... maybe Baars 2008.

R: Indeed, this is a common method. We modified this as follows:

The atmospheric boundary layer height can be determined from LIDAR by using the Haar wavelet transform (HWT) method (Baars et al. 2008). Furthermore, the boundary layer height was retrieved from vertical profiles of potential temperature by using the gradient method (Seidel et al. 2010, Li et al. 2021).

10. p5, l131: p7, l176: You are able to measure ozone as well... should be mentioned.

R: We listed all the parameters that the balloon measured in Table 2. And we added the following description (ozone meaurement) in p7, l176.

Besides, a radiosonde balloon which was operated by the Institute of Energy and Climate Research - Stratosphere (IEK-7) measured the atmospheric parameters from ground to 25 km altitude. COBALD was part of a CFH / ECC ozone / RS41 payload to provide the backscatter coefficients as well as air temperature, air pressure, relative humidity, wind, and ozone concentration with the temporal and spatial resolution being 1s and about 5 m vertically.

11. p7, l183: The COBALD sonde is used to validate lidar observations. To my opinion, the opposite is always the case. And comparisons of lidar with COLBAD observations remain difficult in the case of large, non-spherical particles such as cirrus ad PSC particles and even coarse dust particles because the FOV of the COBALD system is 6°, the lidar FOV is 0.1°. The lidar measures exactly backscatter (or 180° scattering properties), the COLBALD sonde is not able to do that. The scattering phase function can change a lot over 3 degress around 180° in the case of dust and cirrus particles (non spherical particles).

R: Thank you for mentioning that. We are aware of the potential impact of non-spherical particles on the comparison. The lidar measurements showed low depolarization ratios, which indicates spherical particles. Therefore, we conducted a FOV correction as proposed by Brunamonti et al. (2021) employing mie scattering theory. We have added the following sentence to point this out:

The low depolarization ratios support our assumption that the particles are spherical and that we can use Mie calculations for the FOV correction.

12. Figure 2: The legend in Figure 2 is confusing. Please state clearly what is shown in Figure 2.

R: We have simplified Figure 2 to avoid misunderstandings.

[Figure]

Figure 2: Time series of ground-level extinction coefficients retrieved from LIDAR measurements (both elastic and Raman methods) and from Mie calculations based on OPC raw size distributions as well as size distributions corrected by counting efficiency from February $5^{th}$ to March $5^{th}$, 2018 in downtown Stuttgart.

13. p9, l201: ...Fidas200 underestimates.... and at the end ... you believe that these in situ observation validate the lidar measurements?

R: Indeed this requires clarification. Initially, we say that "This figure (Figure 2) shows that the extinction coefficients from Mie calculations based on raw size distributions are substantially lower than those from LIDAR retrievals by a factor of 4.70 ± 1.49." . Then we give the reason for this underestimation. It is the particle number under-counting between 0.25 μm and 0.5 μm by the Fidas200 OPC e.g. when compared with SMPS data as shown in Figure S1. Based on this comparison we could correct the aerosol size distributions. Using the corrected OPC size distributions, the calculated extinction coefficients show good agreement with LIDAR retrievals. Therefore, we modified the text related to figure 2 as follows:

Figure 2 shows that the extinction coefficients from Mie calculations based on raw OPC size distributions are systematically lower than those from LIDAR retrievals by a factor of 4.70 ± 1.49. The reason for this phenomenon is that the Fidas200 underestimates the particle number by a factor of 2-10 at diameters between 0.25 μm and 0.5 μm when compared with SMPS data as shown in Figure S1. The left side of Figure S1 shows the number size distribution from Fidas200 and the merged size distribution from SMPS and APS measurements. The right plot of Figure S1 shows the accumulated extinction coefficients calculated from Mie theory based on those two size distributions, which shows the substantial difference by a factor of four. Hence, we conclude that the underestimation of particle numbers from 0.25 μm to 0.5 μm is one of the main reasons for the underestimation of extinction coefficients based on uncorrected OPC data.

Based on systematic laboratory measurements with the different particle sizers Fidas200 OPC, SMPS, and APS the FIDAS200 counting efficiency was determined (see Figure S2). This counting efficiency was used to correct all measured size distributions. The corrected size distributions were used to calculate the extinction coefficients via Mie calculation. The time series of the extinction coefficients calculated from the corrected size distribution is shown in Figure 2 (orange area). The calculated extinction coefficients show a reasonable agreement with LIDAR retrievals.

14. Figure 3: the use of container humidity and ambient humidity in the correction of in situ aerosol properties is rather confusing... I have the feeling the in situ observations are far away from being trustworthy.

R: We clarified this text. Since we used no dryer in this campaign the Fidas200 OPC directly measured the ambient aerosol. Hence a humidity correction is unnecessary. We modified the discussion related to humidity correction and Figure 2 to make this clear.

The correlation plot between the extinction coefficient from Fidas 200 and the LIDAR-derived extinction coefficient is shown in Figure 3, which shows a slope and a Pearson correlation coefficient of 1.037 ± 0.015 and 0.878, respectively. As shown in Figure 2 the extinction coefficients retrieved from LIDAR measurement show a similar trend with those calculated based on corrected Fidas 200 size distributions. Please note, that the extinction coefficient based on Fidas 200 data are still a little lower than those based on lidar measurements. This may be caused by a partial loss of water from the aerosol particles due to higher temperatures inside the container. However, the aerosol particles are not expected to reach equilibrium within the residence time of 3 seconds in the sampling line inside the warm container. Please note, that there was no dryer in the sampling line.

15. p9, Figure 4: In these modern times with lidars producing data with 7.5 m vertical resolution and 10 s temporal resolution, the color plot is rather poor.

R: Please note that the lidar data in this plot are calculated based on scanning measurements and hence with a time resolution of 11 minutes and significantly less averaging than typically possible for continuously operated only vertically pointing lidar systems. Hence, the data quality in is reasonable well and suitable for a comparison withe the drone measurements.

16. p11, l250 – p12, l264: The paragraph is trivial. I would skip it. What is new? ... should be always the driving question. PBL determination with lidar and comparison with radiosondes and other approaches was often presented during the past 30 years.

R: We think it is relevant to discuss the comparison of scanning lidar results and drone measurements which have become much more frequent recently.

17. Figure 5: Again, a poor color plot because of the rather poor resolution!

R: We decided to shift Figure 5 to the supplement in order to achieve a better focus on the new findings in the main text.

18. Figure 6: The agreement is reasonable. In situ observation with UAVs are just snaphots. Lidar observations seem to be more representative. What is the signal averaging period? Please state!

R: The averaging time for the lidar system is 11 minutes from 08:14 - 08: 25 on July 09, 2018. We have added the following information to the caption of figure 6 (labelled as figure 5 in current version).

...averaged from 08:14 - 08:25),...

19. Figure 7: The correlation is rather poor. The UAV observations from 9 July are much too high, and the ones measured on 12 July are much too low. The quality of the in situ observations is therefore rather low. What do we learn from such a poor result?

R: Indeed the quality of this in situ observation with the OPC-N3 on the UAV is rather poor e.g. compared to that by the corrected Fidas200 OPC data. This demonstrates what can be the result when flying a low cost sensor on an UAV. We think it is useful to present this as it means that we must be careful with the quality and the operation of *in-situ* measurements. We have modified the corresponding text as follows:

However, the backscatter coefficients retrieved from the UAV-borne OPC in certain UAV flights still show a relatively large deviation from LIDAR retrievals. One reason for this variability is that the UAV cruising speed may affect aerosol sampling by the OPC-N3. The sample were collected perpendicular to the flight direction into the OPC, so we can expect size-dependent discrimination of larger particles. Compared to the Fidas200 OPC as shown in section 3.1, the OPC-N3 data show a significantly higher variability. This means that we must be careful with the quality and the operation of such *in-situ* measurements especially when no reference data like lidar are available.

20. p16, l304: I asked myself, why do the authors want to validate lidar observations of the backscatter coefficient? Ground based lidars (and airborne lidars) are used since the 1970s and space lidars since the 1990s. Do we still need to validate lidars? That makes sence. But why do we need to validate ground-based lidars? Please explain, what is so critical with lidar backscatter observations?

R: Indeed, ground based lidars (and airborne lidars) have been used since decades. However, gaps still exist when comparing scanning lidars with *in-situ* measurements and model simulations in terms of backscatter coefficients. As stated by Hoshyaripour et al. (2019) and Zhang et al. (2022), e.g. inappropriate particle shape parameterisations can cause discrepancies of backscatter coefficients by a factor of 3-4, which demonstrates how crucial it is to use an appropriate parameterisations for e.g. dust particles. In this manuscript, we want to show the factors that potentially affect a backscatter coefficient comparison e.g. when converting aerosol particle size distributions to aerosol backscatter coefficients. This should help us to better parameterise or model aerosol optical proprieties in further model development. We have modified the corresponding text as follows:

However, to our best knowledge, so far no dedicated comparison of scanning LIDAR measurement with *in-situ* observation has been performed over a wide altitude range and over such long period for comparison at ground level (e.g. one month dataset with 10 minute resolution). Also in order to bridge the gaps that are often encountered between remote sensing and *in-situ* observations, we compared datasets on aerosol spatial-temporal distributions and evolution combining remote sensing and *in-situ* measurements.

21. Figure 8: Again, the color plot is very poor because the vertical and temporal resolution is so poor. What does the color show in (b), probably distance, is not given in the figure.

R: The temporal resolution for the vertical point data is 1 minute. We show continuous measurements for two hours only vertically pointing in this case. The color shown in (b) is the radiosonde flight altitude and we have added a legend to explain this.

22. p18, l341: Again, I think the opposite is true. The lidar does not need validation after 40 years of consistent, high quality aerosol observations. Moreover, the lidar shows (in Figure 10) that even a balloonborne backscatter sonde (one snapshot-like measurement per height) can be used to characterize the height profile of aerosols. The advantage of lidar, on the other hand, is the potential to monitor aerosol developments of the vertical aerosol distribution over long time periods, continuously.

R: The comparison between the Raman lidar and the backscatter sonde shows how good an agreement between remote sensing and in-situ measurement can be if proper data analysis (e.g. displacement correction etc.) is done. Furthermore, it demonstrates how complementary the different methods are. The *in-situ* measured atmospheric parameters can significantly improve the lidar retrievals e.g. for cases when no Raman measurements are possible. Furthermore, a combination of the backscatter coefficients for the three different wavelengths gives access to wavelength dependencies.

This comparison highlights the complementary advantages of lidar's continuous measurement capability and COBALD *in-situ* two wavelength data for characterising aerosol particles from near ground level up to the

[Figure]

Figure 7: Time series of range corrected LIDAR signal and radiosonde vertical trajectory (white dash line) (a) and horizontal displacement of the balloon during this experiment (b) on July $12^{th}$, 2018 at Jülich research center.

stratosphere.

**References**

Adam, M., Nicolae, D., Stachlewska, I. S., Papayannis, A. & Balis, D. (2020), 'Biomass burning events measured by lidars in earlinet – part 1: Data analysis methodology', *Atmospheric Chemistry and Physics* **20**(22), 13905–13927.
**URL:** *https://acp.copernicus.org/articles/20/13905/2020/*

Baars, H., Ansmann, A., Engelmann, R. & Althausen, D. (2008), 'Continuous monitoring of the boundary-layer top with lidar', *Atmospheric Chemistry and Physics* **8**(23), 7281–7296.
**URL:** *https://acp.copernicus.org/articles/8/7281/2008/*

Baars, H., Kanitz, T., Engelmann, R., Althausen, D., Heese, B., Komppula, M., Preißler, J., Tesche, M., Ansmann, A., Wandinger, U., Lim, J.-H., Ahn, J. Y., Stachlewska, I. S., Amiridis, V., Marinou, E., Seifert, P., Hofer, J., Skupin, A., Schneider, F., Bohlmann, S., Foth, A., Bley, S., Pfüller, A., Giannakaki, E., Lihavainen, H., Viisanen, Y., Hooda, R. K., Pereira, S. N., Bortoli, D., Wagner, F., Mattis, I., Janicka, L., Markowicz, K. M., Achtert, P., Artaxo, P., Pauliquevis, T., Souza, R. A. F., Sharma, V. P., van Zyl, P. G., Beukes, J. P., Sun, J., Rohwer, E. G., Deng, R., Mamouri, R.-E. & Zamorano, F. (2016), 'An overview of the first decade of polly$^{NET}$: an emerging
network of automated raman-polarization lidars for
continuous aerosol profiling', *Atmospheric Chemistry and Physics* **16**(8), 5111–5137.
**URL:** *https://acp.copernicus.org/articles/16/5111/2016/*

Brunamonti, S., Martucci, G., Romanens, G., Poltera, Y., Wienhold, F. G., Hervo, M., Haefele, A. & Navas-Guzmán, F. (2021), 'Validation of aerosol backscatter profiles from raman lidar and ceilometer using balloon-borne measurements', *Atmospheric Chemistry and Physics* **21**(3), 2267–2285.
**URL:** *https://acp.copernicus.org/articles/21/2267/2021/*

Burton, S. P., Ferrare, R. A., Hostetler, C. A., Hair, J. W., Rogers, R. R., Obland, M. D., Butler, C. F., Cook, A. L., Harper, D. B. & Froyd, K. D. (2012), 'Aerosol classification using airborne high spectral resolution lidar measurements – methodology and examples', *Atmospheric Measurement Techniques* **5**(1), 73–98.
**URL:** *https://amt.copernicus.org/articles/5/73/2012/*

Burton, S. P., Vaughan, M. A., Ferrare, R. A. & Hostetler, C. A. (2014), 'Separating mixtures of aerosol types in airborne high spectral resolution lidar data', *Atmospheric Measurement Techniques* **7**(2), 419–436.
**URL:** *https://amt.copernicus.org/articles/7/419/2014/*

Fernald, F. G. (1984), 'Analysis of atmospheric lidar observations: some comments', *Appl. Opt* **23**(5), 652–653.

Groß, S., Esselborn, M., Weinzierl, B., Wirth, M., Fix, A. & Petzold, A. (2013), 'Aerosol classification by airborne high spectral resolution lidar observations', *Atmospheric Chemistry and Physics* **13**(5), 2487–2505.
**URL:** *https://acp.copernicus.org/articles/13/2487/2013/*

Groß, S., Freudenthaler, V., Schepanski, K., Toledano, C., Schäfler, A., Ansmann, A. & Weinzierl, B. (2015), 'Optical properties of long-range transported saharan dust over barbados as measured by dual-wavelength depolarization raman lidar measurements', *Atmospheric Chemistry and Physics* **15**(19), 11067–11080.
**URL:** *https://acp.copernicus.org/articles/15/11067/2015/*

Hoshyaripour, G., Bachmann, V., Förstner, J., Steiner, A., Vogel, H., Wagner, F., Walter, C. & Vogel, B. (2019), 'Effects of particle nonsphericity on dust optical properties in a forecast system: Implications for model-observation comparison', *J. Geophys. Res. Atmos.* **124**(13), 7164–7178.

Hu, Q., Goloub, P., Veselovskii, I. & Podvin, T. (2022), 'The characterization of long-range transported north american biomass burning plumes: what can a multi-wavelength mie–raman-polarization-fluorescence lidar provide?', *Atmospheric Chemistry and Physics* **22**(8), 5399–5414.
**URL:** *https://acp.copernicus.org/articles/22/5399/2022/*

Ke, J., Sun, Y., Dong, C., Zhang, X., Wang, Z., Lyu, L., Zhu, W., Ansmann, A., Su, L., Bu, L. et al. (2022), 'Development of china's first space-borne aerosol-cloud high-spectral-resolution lidar: retrieval algorithm and airborne demonstration', *PhotoniX* **3**(1), 17.

Klett, J. D. (1985), 'Lidar inversion with variable backscatter/extinction ratios', *Appl. Opt* **24**(11), 1638–1643.

Li, H., Liu, B., Ma, X., Jin, S., Ma, Y., Zhao, Y. & Gong, W. (2021), 'Evaluation of retrieval methods for planetary boundary layer height based on radiosonde data', *Atmospheric Measurement Techniques* **14**(9), 5977–5986.

Liu, Z., Matsui, I. & Sugimoto, N. (1999), 'High-spectral-resolution lidar using an iodine absorption filter for atmospheric measurements', *Optical Engineering* **38**(10), 1661–1670.

Mylonaki, M., Giannakaki, E., Papayannis, A., Papanikolaou, C.-A., Komppula, M., Nicolae, D., Papagiannopoulos, N., Amodeo, A., Baars, H. & Soupiona, O. (2021), 'Aerosol type classification analysis using earlinet multiwavelength and depolarization lidar observations', *Atmospheric Chemistry and Physics* **21**(3), 2211–2227.
**URL:** *https://acp.copernicus.org/articles/21/2211/2021/*

Piironen, P. & Eloranta, E. (1994), 'Demonstration of a high-spectral-resolution lidar based on an iodine absorption filter', *Opt. Lett.* **19**(3), 234–236.

Schillinger, M., Morancais, D., Fabre, F. & Culoma, A. J. (2003), ALADIN: the lidar instrument for the AEOLUS mission, *in* H. Fujisada, J. B. Lurie, M. L. Aten, K. Weber, J. B. Lurie, M. L. Aten & K. Weber, eds, 'Sensors, Systems, and Next-Generation Satellites VI', Vol. 4881, International Society for Optics and Photonics, SPIE, pp. 40 – 51.
**URL:** *https://doi.org/10.1117/12.463024*

Seidel, D. J., Ao, C. O. & Li, K. (2010), 'Estimating climatological planetary boundary layer heights from radiosonde observations: Comparison of methods and uncertainty analysis', *Journal of Geophysical Research: Atmospheres* **115**(D16).

Shin, J., Kim, G., Kim, D., Tesche, M., Park, G. & Noh, Y. (2024), 'Multi-section reference value for the analysis of horizontally scanning aerosol lidar observations', *Atmospheric Measurement Techniques* **17**(2), 397–406.
**URL:** *https://amt.copernicus.org/articles/17/397/2024/*

Wandinger, U. (2005), Raman lidar, *in* 'Lidar', Springer, pp. 241–271.

Wehr, T., Kubota, T., Tzeremes, G., Wallace, K., Nakatsuka, H., Ohno, Y., Koopman, R., Rusli, S., Kikuchi, M., Eisinger, M., Tanaka, T., Taga, M., Deghaye, P., Tomita, E. & Bernaerts, D. (2023), 'The earthcare mission – science and system overview', *Atmospheric Measurement Techniques* **16**(15), 3581–3608.
**URL:** *https://amt.copernicus.org/articles/16/3581/2023/*

Zhang, H., Wagner, F., Saathoff, H., Vogel, H., Hoshyaripour, G., Bachmann, V., Förstner, J. & Leisner, T. (2022), 'Comparison of scanning lidar with other remote sensing measurements and transport model predictions for a saharan dust case', *Remote Sensing* **14**(7).
**URL:** *https://www.mdpi.com/2072-4292/14/7/1693*

Zhang, M., Tian, P., Zeng, H., Wang, L., Liang, J., Cao, X. & Zhang, L. (2021), 'A comparison of wintertime atmospheric boundary layer heights determined by tethered balloon soundings and lidar at the site of sacol', *Remote Sensing* **13**(9), 1781.

**Reply to the referee's comments**

We thank the referee for the useful comments, which helped us to improve the quality of our manuscript.

In the following, the referees' comments are given in black.

Our point-to-point replies are marked by "R" and are in blue.

Changes to the manuscript text are in green.

**Comments by referee #2**

The manuscript compares the aerosol optical properties and boundary layer height retrieved from a scanning lidar with various in-situ measurements, including ground-based, UAV, and balloon measurements. This comparison to some extent validates the feasibility of scanning lidars and new vertical in-situ measurement techniques, and also identifies their calibration methods. The research content aligns with the publication scope of the Atmospheric Measurement Techniques journal, and publication is recommended after revisions.

**Specific Comments:**

1) The English grammar and structure of the manuscript need further improvement.

R: Thank you for taking the time to review our manuscript and provide feedback on its grammar and structure. We will carefully consider your suggestions and work on improving the manuscript to ensure it meets the highest standards.

2) It is difficult to discern the research purpose the authors intend to convey from the manuscript. If the aim of the study is to validate and analyze the errors of the lidar retrieval using in-situ measurements, it would be more appropriate to calibrate and correct the in-situ measurements first. Then, the corrected results should be compared with the lidar retrieval results for a more reliable comparison, rather than initially comparing the lidar retrieval results with the uncorrected in-situ measurements.

R: We generally agree but we think it is useful for the readers to learn also about the calibration and comparison procedures. Actually, our OPC (Fidas 200) was initially calibrated by reference particles according to the operation manual. However, our comparisons with other particle sizers revealed that the Fidas200 still underestimated the aerosol particle number in the range from 0.25 $\mu$m to 0.5 $\mu$m. This underestimation may partially be related to aerosol particle properties. For example, some particles at such small size were not detected by Fidas200 e.g. due to lower scattering cross sections but were detected by the scanning mobility particle sizer SMPS. Also the OPC-N3 flown on the UAV was calibrated before measurement. However, the sample were collected perpendicular to the flight direction into the OPC, so we can expect a size-dependent discrimination of larger particles. Compared to the Fidas200 OPC as shown in section 3.1, the OPC-N3 shows a higher variability compared to lidar measurements. Therefore, we should be careful with the quality and the operation of such *in-situ* measurements especially if no reference instruments are available. The balloon borne backscatter sonde shows a quite good agreement with the continuous lidar measurements after applying reasonable correction for wavelength, field of view, and spatial displacement. The results presented in this manuscript show that not only the accuracy of the instruments but also the operation and the corresponding meteorological factors need to be considered for these comparisons. We have modified the text related to the comparison with the in-situ measurements as follows:

However, the backscatter coefficients retrieved from the UAV-borne OPC in certain UAV flights still show a relatively large deviation from LIDAR retrievals. One reason for this variability is that the UAV cruising speed may affect aerosol sampling by the OPC-N3. The sample were collected perpendicular to the flight direction into the OPC, so we can expect size-dependent discrimination of larger particles. Compared to the Fidas200 OPC as shown in section 3.1, the OPC-N3 data show a significantly higher variability. This means that we must be careful with the quality and the operation of such *in-situ* measurements especially when no reference data like lidar are available.
This comparison highlights the complementary advantages of lidar's continuous measurement capability and

COBALD *in-situ* two wavelength data for characterising aerosol particles from near ground level up to the stratosphere.

3) The analysis and explanation of the comparison results in Chapter 3.1 are confusing. Lidar detects aerosols in ambient conditions, so why is indoor humidity being incorporated into the Mie calculation for comparison? It seems the authors intend to convey that the in-situ measured aerosols are not completely dry, but when incorporated into the Mie calculation, they are treated as if they are in a dry state, leading to higher results. This could serve as a potential explanation for the differences in the comparison results, but it still does not clarify why indoor humidity is being used in the calculation.

R: This is indeed confusing. Actually, there was no dryer in the sampling line of the Fidas200 during this campaign. Hence the Fidas directly measured the ambient aerosol. Hence the humidity correction is unnecessary. Consequently, we modified the discussion related to humidity correction as follows:

The correlation plot between the extinction coefficient from Fidas 200 and the LIDAR-derived extinction coefficient is shown in Figure 3, which shows a slope and a Pearson correlation coefficient of $1.037 \pm 0.015$ and 0.878, respectively. As shown in these two figures, the extinction coefficients retrieved from LIDAR measurement show a similar trend with Fidas 200 retrieval after Fidas 200 size correction. In addition, we also found that the Fidas 200 retrieval was still a bit lower than the lidar measurement due to the fact that the aerosol particles lost their water partly inside the container but did not reach equilibrium within the 3 s residence time in the sampling line. Please note, that there was no dryer in the sampling line.

4) Figures 4 and 5 should be redrawn. It is inappropriate to directly plot potential temperature profiles while the x-axis represents time. Instead, the measured potential temperature profiles should first be used to retrieve the boundary layer height based on drone detection, and then labeled on the graph for comparison with boundary layer heights from other sources.

R: Thank you for pointing to this. We have analysed the potential temperatures for boundary layer heights according to the gradient method and plotted the values for comparison with the other data in figure 4. To better focus the manuscript on the new findings we moved figure 5 to the supplement.

[Figure]

Figure 4: Time series of backscattering coefficients (contour), boundary layer heights (PBLH, pink squares) and residual layer heights (RLH, yellow squares) retrieved from scanning LIDAR, as well as boundary layer heights obtained from UAV measurements (black star with white circle surrounded) and from ERA5 dataset (white dashed line) measured by UAV on July $9^{th}$, 2018.

5) The experimental effort is large and that aspect deserves publication. However, the results are not new. Lidar comparisons with COBALD observations and with in situ aerosol observations have already been performed and published. The use of a scanning lidar in these comparisons may be a new aspect.

R: We agree that several comparison studies with lidar measurements have been conducted previously. However, this is not the case for the scanning Raman lidar that we used in this study. Given the special capabilities

of the scanning lidar we think it is valuable for the readers to learn how it can be compared to other methods.

1. The scanning lidar allows to conduct horizontal measurements which facilitates direct comparison with ground level observations (Shin et al. 2024). This direct comparison with ground level observations allows us to do long-term comparisons with high time resolution (e.g 10-minute resolution over one month), which facilitates a robust statistical evaluation of this kind of comparison. Furthermore, employing multi-angle analysis of an aerosol layer allows to retrieve backscatter coefficients independently without needing to assume a lidar ratio (Zhang et al. 2021). Since the scanning lidar retrieval algorithm isn't very common we think it is justified to demonstrate its quality experimentally.

2. As far as we know, this manuscript firstly evaluates the comparison between scanning lidar and *in-situ* measurements over a wide altitude range from ground level to the stratosphere (around 20 km). This comparison over a wide altitude range includes the high variability of aerosol concentrations within the boundary layer and relatively stable concentrations in free troposphere, which shows the reader the relevant time-scales and spatial-scales of aerosol evolution when conducting such comparison.

3. Compared with previous literature, this manuscript not only focuses on the comparison of remote sensing and *in-situ* measurements but also discusses the potential meteorological factors that affect the comparison results (section 3.3). We have modified the end on the introduction as follows:

However, to our best knowledge, so far no dedicated comparison of scanning LIDAR measurement with *in-situ* observation has been performed over a wide altitude range and over such a long time period for comparison at ground level (e.g. one month dataset with 10 minute resolution). Also in order to bridge the gaps that are often encountered between remote sensing and *in-situ* observation, we compared datasets on aerosol spatial-temporal distributions and evolution combining remote sensing and *in-situ* measurements.

6) Several "scatter" in the manuscript should be "scattering", like scattering coefficient, single scattering albedo, etc.

R: We have check the manuscript carefully and corrected this.

8) Line 27: "the" should be removed.

R: We have removed "the"

7) "LIDAR" is a common way to write lidar, so the author's choice to use it to refer to lidar is fine. However, when introducing terms such as CALIPSO and EARLINET, their official writing "Lidar" should be used.

R: We have checked the spelling carefully in this manuscript to keep this term consistent in this manuscript and also with previous literature.

8) Line 31: "and" should be "of".

R: We have changed "and" to "of".

**References**

Shin, J., Kim, G., Kim, D., Tesche, M., Park, G. & Noh, Y. (2024), 'Multi-section reference value for the analysis of horizontally scanning aerosol lidar observations', *Atmospheric Measurement Techniques* **17**(2), 397–406.
**URL:** *https://amt.copernicus.org/articles/17/397/2024/*

Zhang, M., Tian, P., Zeng, H., Wang, L., Liang, J., Cao, X. & Zhang, L. (2021), 'A comparison of wintertime atmospheric boundary layer heights determined by tethered balloon soundings and lidar at the site of sacol', *Remote Sensing* **13**(9), 1781.

---

## Author Response (AR2)

**Reply to the referee's comments**

We thank the editor and referees for the useful comments, which helped us to improve the quality of our manuscript.

In the following, the referees' comments are given in black.

Our point-to-point replies are marked by "R" and are in blue.

Changes to the manuscript text are in green.

1. Before we can make the final acceptance, I would like you to make some edits to the use of 'backscatter' and 'backscattering'. I checked the use with the two referees and had conflicting advice. Based on a combination of the referee's recommendations and the use of these terms in Chapter 10 of 'Aerosols and Climate' (Ed Carlaw), I would like you to use the term 'scattering' or 'backscattering' when referring to the process, but use 'backscatter' when in conjunction with 'coefficient'.

   R: We have check the manuscript carefully and corrected this.

2. Regarding figure 8: please ensure that the colour schemes used in your maps and charts allow readers with colour vision deficiencies to correctly interpret your findings. Please check your figures using the Coblis – Color Blindness Simulator (https://www.color-blindness.com/coblis-color-blindness-simulator/) and revise the colour schemes accordingly.

   R: We have changed the colormap accordingly to ensure the colour schemes allow readers with colour vision deficiencies to correctly interpret the findings .

[Figure]

Figure 1: Backscatter coefficients measured by balloon-borne COBALD and LIDAR (left) as well as aerosol volume and particle depolarization ratio measured by LIDAR (right) on the night time of July $12^{th}$, 2018 at Jülich research center. (The integration time of the LIDAR data is 1 hour from 21:19 to 22:19.)